# Effect of Salt Bath Nitriding and Reoxidation Composite Texture on Frictional Properties of Valve Steel 4Cr10Si2Mo

Yifan Dai [1], Zefei Tan [1,*], Wengang Chen [1,*], Dongyang Li [2], Jubang Zhang [1], Zexiao Wang [1], Yukun Mao [1], Yuhao Wang [1] and Wenxuan Guo [1]

1    College of Mechanical and Transportation, Southwest Forestry University, Kunming 650224, China
2    Department of Chemical and Materials Engineering, University of Alberta, Edmonton, AB T6G 2H5, Canada
*    Correspondence: tanzefei999@swfu.edu.cn (Z.T.); chenwengang999@swfu.edu.cn (W.C.)

**Abstract:** In order to improve the service life of 4Cr10Si2Mo valve steel, laser processing technology was used to prepare triangular textures with different area occupancies on the surface of 4Cr10Si2Mo, and then 4Cr10Si2Mo was subjected to salt bath nitridation (salt bath temperature 580 °C) and oxidation (oxidation temperature 400 °C). The mechanism of composite surface treatment technology on friction and wear performance of valve steel was explored. The effect of triangular texture on working surface stress and hydrodynamic pressure was explored using simulation technology, and the mechanism of texture in friction was further studied. The XRD test results showed that after salt bath nitriding and reoxidation treatment, the surface of 4Cr10Si2Mo mainly contained $Fe_2N$ oxide film and $Fe_3O_4$ and other components. The XPS test showed that the nitriding layer contained Cr-N, and the surface hardness reached 710.5 $HV_{0.5}$. The simulation results showed that introducing surface texture will increase the stress on the contact surface, especially near the texture. However, under lubricating conditions, the flow field in the textured lumen produces a wedge effect, which increases the oil film pressure. After salt bath nitriding composite texture treatment, the wear resistance of 4Cr10Si2Mo significantly improved under the synergistic effect of the nitrided layer dominated by the magnetite phase ($Fe_3O_4$) and the microtexture. $Fe_3O_4$ can reduce the friction coefficient and resist oxidation reactions. In addition, the texture of the area occupancy of the texture also affects the surface tribological properties. The texture with an area occupancy rate of 11.45% (low × high is 0.3 mm × 0.3 mm) had the best anti-friction effect, and the friction coefficient reduced by 65%.

**Keywords:** triangular texture; QPQ salt bath nitriding; friction coefficient; wear mechanism

## 1. Introduction

4Cr10Si2Mo is martensitic heat-resistant steel, which is often used in the manufacture of exhaust valves of automobile engines or intake valve stems of low-power engines. As martensitic steel, 4Cr10Si2Mo is generally treated by quenching and high-temperature tempering to increase its strength. 4Cr10Si2Mo contains Si and Mo elements, which makes it resistant to high temperature and oxidation, and its microstructure and properties are also stable. The rapid development of the industry has improved the performance of automobiles and the popularization of turbocharging systems, increasing the engine's temperature as a whole. In order to meet current needs, it is of great significance to improve the tribological properties of 4Cr10Si2Mo valve steel.

At present, there are different ways to reduce the friction and wear of friction pairs, mainly in advanced materials [1–3], surface modification [4,5], lubricants [6], etc. Surface texture as an environmentally friendly and biocompatible method in surface modification has received extensive attention in the friction field [7,8]. Surface texture is the use of specific processing techniques to create pits, grooves, or raised structures on the surface of an object by physical or chemical means. Commonly used processing techniques are EDM, 3D printing, laser processing, shot peening, LIGA (lithography, electroforming, injection

moulding), chemical reactive ion etching, and nanoimprinting [9,10]. Among them, the advantages of laser processing, such as a wide range of materials, fast processing speed, and no pollution to the environment, are widely known. The primary function of texture is the effect of fluid pressure and the storage of wear debris, reducing the wear on the surface of the material and thereby reducing the coefficient of friction. At present, the focus of texture research is mainly on the influence of texture shape, size, depth, density, arrangement, and friction conditions on the tribological properties of materials. Conventional shapes such as circles, grooves, and rectangles are the most studied tribological geometry textures that can improve the coefficient of friction and wear. Later, researchers explored the effect of unconventional texture on tribological properties. Uddin [11] compared the effects of circular, rectangular, and elliptical textures on the friction pair under lubrication conditions and found that the rectangular texture has the best anti-friction effect. Zhang [12] compared the friction reduction effect of circular texture with bullet-shaped and fish-shaped texture, and the friction-reducing effect of bullet-shaped and fish-shaped bright texture was prominent. Maldonado-Cortés [13] conducted a comparative study of circles, triangles, lines, squares, crosses, and "S" shapes. They showed that at low contact stress, all textures reduced the friction coefficient, while closed geometry shapes (circle, triangle, square) showed less wear. Chen [14] studied the hydrodynamic properties of textures with irregular and regular arrangements. The study showed that the hydrodynamic properties of textures at the same depth and area occupancy were affected by the texture arrangement, especially when the textures were different. In the regular arrangement, the synergistic effect of texture is more prominent.

At present, the nitriding methods mainly include laser nitriding [15–17], plasma nitriding [18], salt bath nitriding [19,20], and other technologies. Salt bath nitriding seems to be the default in the automotive industry as the primary technology to increase the surface strength and corrosion resistance of parts. Compared to other nitriding technologies, salt bath nitriding has apparent advantages in terms of cost and efficiency. The use of cyanate eliminates wastewater treatment operations. It solves environmental problems. M. Mamatha Gandhi [21] studied the dry sliding friction and wear of AISI 304; the formed nitride can increase the material's surface hardness, and the nitrided layer's hardness reached 835Hv. Wong-Ángel W D [22] carried out salt bath nitriding of DIN-16MnCr5 followed by oxidation, and the results showed that the wear and corrosion resistance was improved fivefold. Chen [23] explored the fretting wear of 2.25Cr-1Mo steel in liquid sodium at different salt bath temperatures (550 °C and 590 °C), and the friction reduction effect was the best when the temperature was 590 °C.

With the development of computer technology, finite element analysis of objects with computer-aided tools (CAE) has been widely used. Hegadekatte [24] used the incremental global model developed by the Archard wear formula to simulate the wear of the planetary gear friction pair. The results show that the wear at the pitch line of the gear is the smallest, and the wear at the part below the pitch line is the most significant. Xing Y [25] analyzed the texture and smooth surface by the finite element method (FEM) and found that the surface texture can improve the stress distribution of the interface. Yan [26] established a two-dimensional model of particles between the two surfaces using the finite element software Abaqus to establish two surface shear planes. According to the stress cloud diagram, the variation of stress and friction of particles with different parameters was found, and particles' motility and shear expansion during friction were obtained. Li [27] used Fluent software to simulate and analyze the herringbone, circular, groove, and composite textures to explore different textures' dynamic pressure lubrication characteristics. The simulation results show that the herringbone texture has a strong bearing capacity. The average dynamic pressure value of the circular texture is considerable. The groove-shaped texture can improve the flow rate of lubricating oil, but the bearing capacity is weak. The composite texture ensures a specific bearing capacity while ensuring the fluidity of the lubricating oil. Finite element analysis can save much time by performing some experiments on the computer to obtain data that are difficult to obtain. However, the problem of CAE

simulation accuracy has always been controversial, and it is worth further studying to solve the problem of finite element accuracy.

Many studies have proved that the nitrided layer has high hardness and can improve wear and corrosion resistance, but it can easily fail due to its high brittleness. In this work, to increase the service life of 4Cr10Si2Mo valve steel, oxidation was carried out after nitriding, and a nitride layer and an oxide layer were produced simultaneously. On this basis, laser technology fabricated micro-pits on the surface to form a nitride–oxide layer microtexture. The friction and wear experiments were carried out using a self-lubricating friction pair and 4Cr10Si2Mo valve steel under full lubrication conditions. Abaqus software was used to explore the effect of texture addition on the surface stress distribution. Fluent software analyzed the influence of textures with different area occupancy on the bearing capacity of the oil film. The effect of nitriding composite textured surfaces on tribological properties was examined by combining experimental and simulation results.

## 2. Materials and Experiments

### 2.1. Material Preparation

The upper sample material used in the experiment was gray cast iron HT200 (Hengrongchang Metal Materials Co, Kunshan, China). HT200 is a self-lubricating material with steel as the matrix and a large amount of graphite. The lower sample material used in the test was 4Cr10Si2Mo martensitic heat-resistant steel. As shown in Figure 1, it is a pin-disk model in which the diameter of the cylindrical pin is 4.6 mm and the height is 12.7 mm. The diameter of the inner ring of the specimen ring is 38 mm, and the diameter of the outer ring is 54 mm. The chemical compositions of the two materials are shown in Table 1. Triangular micro-textures were etched on the surface of the samples by the FLS-FB50 (Frasers Laser Technology Co, Shenzhen, China) mobile rotary laser marking machine. The marking parameters were as follows: the laser output power was 25 W, the speed was 2000 mm/s, and the number of laser processing times was 5. The specific dimensions of the texture are shown in Table 2. The material was polished with 160-2000 mesh sandpaper on a PG-1S metallographic polishing machine with a 0.5 μm diamond polishing paste, placed in an ultrasonic cleaner for 10 min after polishing, and taken out to dry. The textured surface samples were subjected to salt bath nitridation, and the specific operation steps were as follows: heating → salt bath nitriding → desalting → drying → polishing → preheating → oxidation → desalting → polishing. The nitriding temperature was 580 °C and the time was 4 h; the oxidation temperature was 400 °C and the time was 4 h.

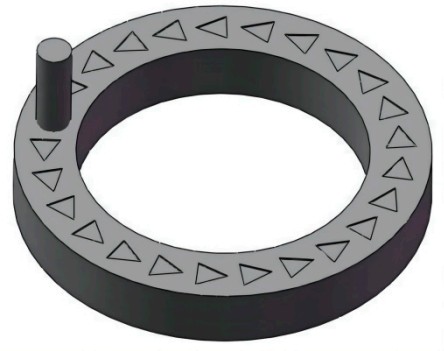

**Figure 1.** Triangular texture diagram.

### 2.2. Friction and Wear Experiments

We used the MMW-1A (Jinan Hengxu Testing Machine Technology Co., Ltd, Jinan, China) vertical universal friction and wear testing machine to conduct the pin-disk friction performance experiment. The movement mode was gyratory movement, and the direction was counterclockwise. The schematic diagram is shown in Figure 2. The experiment was conducted at room temperature, and the lubricants were chemically stable mineral oils.

The upper sample material was HT200, commonly used to manufacture valve guides. The lower sample material was 4Cr10Si2Mo, commonly used to manufacture valve stems. The applied load was 130 N, the rotation speed was 120 r/min, and the test time was 4 h. The test adopted the oil lubrication state.

**Table 1.** The mass fractions of the chemical compositions of the upper and lower samples (mass fraction %).

| Chemical Composition | C | Si | Mn | P | Cr | Ni | Mo | S | Fe |
|---|---|---|---|---|---|---|---|---|---|
| 4Cr10Si2Mo | 0.39 | 2.24 | 0.45 | 0.022 | 9.90 | 0.30 | 0.75 | 0.006 | Bal |
| HT 200 | >3.0 | >1.4 | >0.6 | >0.15 | - | - | - | >0.12 | Bal |

**Table 2.** Experimental group number.

| Non-Nitrided Group | Nitriding Group | Texture Parameters d × h (mm) | Texture Area Density (Ar) |
|---|---|---|---|
| TA | NA | Nitriding 0.3 × 0.3 | 11.45% |
| TB | NB | Nitriding 0.5 × 0.5 | 23.86% |
| TC | NC | Nitriding 0.7 × 0.7 | 38.98% |
| GH | NS | Untextured surface | 0 |

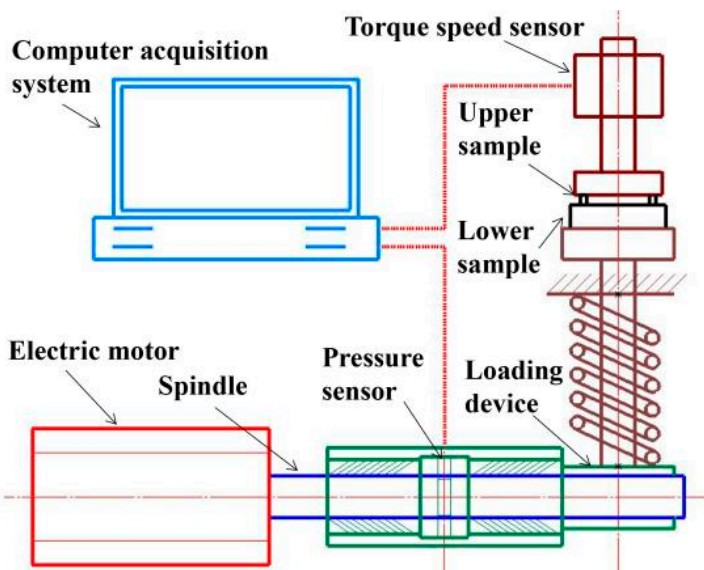

**Figure 2.** Working principle of MMW-1A vertical universal friction and wear testing machine.

The following material characterization instruments were used in the experiment: a multi-functional electronic balance (FA124) to measure the quality of the samples under each friction pair before and after the experiment, and the average value was calculated 3 times each time to calculate the wear amount; a ray diffractometer (Japan Rigaku Corporation D/MAX-UltimaX, Tokyo, Japan) to detect the phase composition of the matrix before and after nitriding; ray photoelectron spectroscopy (Thermo EscaLab Xi + X, Uppsala, Sweden) to analyze the change state of elements before and after nitriding; a Vickers microhardness tester (MH-6L, Hengyi Precision Instrument Co., Ltd., Shanghai, China) to measure the surface hardness of the substrate before and after nitriding; an optical microscope (Jinan Hengxu YM200, Jinan, China) to photograph the surface after wear scars for macro friction and wear analysis; an electron microscope (German Zeiss sigma 500vp, Oberkochen, Germany) to photograph the wear scars; an elemental energy spectrum analyzer (Oxford X-MAX, Oxford, UK) to obtain the relevant element distribution

and mass fraction; a laser confocal microscope (Olympus, Tokyo, Japan) to detect the three-dimensional morphology and surface roughness after wear.

### 2.3. Simulation

Abaqus (2021, Simulia, Dearborn, MI, USA) analyzed the influence of different triangular textures on surface stress. The pin-disk finite element model was established according to the experimental situation. In order to reduce the calculation, a part of it was taken for simulation analysis. Figure 3 is the finite element model mesh. We set the overall size of the entire mesh to 0.1 mm, and refined the mesh at the contact surface. The density of HT200 is 7080 Kg/m$^3$, Young's modulus is 130 GPa, and Poisson's ratio is 0.3. The density of 4Cr10Si2Mo is 7620 Kg/m$^3$, Young's modulus is 206 GPa, and Poisson's ratio is 0.2. Other settings refer to the actual working conditions of the experiment.

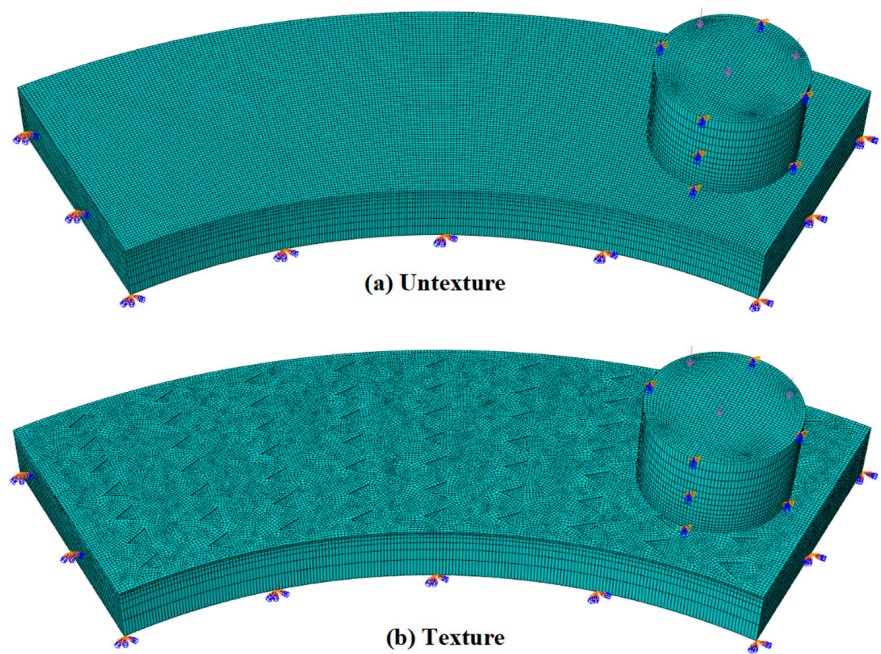

**Figure 3.** Finite element model mesh. (**a**) Untextured surface. (**b**) Textured surface.

Figure 4a is a triangular textured flow field model. Periodic boundaries are set in the y direction; the pressure input surface and the pressure output surface are set in the x direction; the upper wall is set as a moving wall, and the velocity is 0.289 m/s; the lower wall is set as a static wall. The working pressure is the ambient standard atmospheric pressure. Figure 4b shows the oil film parameters, where h$_1$ is the oil film thickness and h$_2$ is the texture depth. Figure 4c shows the specific dimensions of the triangular texture, where d$_0$ is the length of the bottom edge of the texture and h$_0$ is the texture's height. In the calculation, the density of lubricating oil is 801.4 kg/m$^3$, and the dynamic viscosity is 0.0839 kg/(m·s). Generally, the Reynolds number Re determines whether the fluid flow state is laminar or turbulent. When the Reynolds number $Re \leq 2300$, the flow state is laminar; when the Reynolds number $Re \geq 4000$, the flow state is turbulent; when the Reynolds number is between the two, the flow state is a transition state. Formula (1) is the formula for calculating the Reynolds number.

$$R_e = \frac{\rho v d}{\mu} \tag{1}$$

where $P$ is the fluid density, $v$ is the fluid velocity, $d$ is the oil film thickness, and $\mu$ is the absolute viscosity. After calculation, the value of $Re$ was much less than 2300, so this experiment adopted the laminar flow model for simulation calculation. The saturated

vapor pressure of lubricating oil is generally above 20,000 Pa. In this experiment, the sliding speed of the upper wall surface was relatively slow, and the pressure generated was much lower than this value, so cavitation did not occur during the experiment.

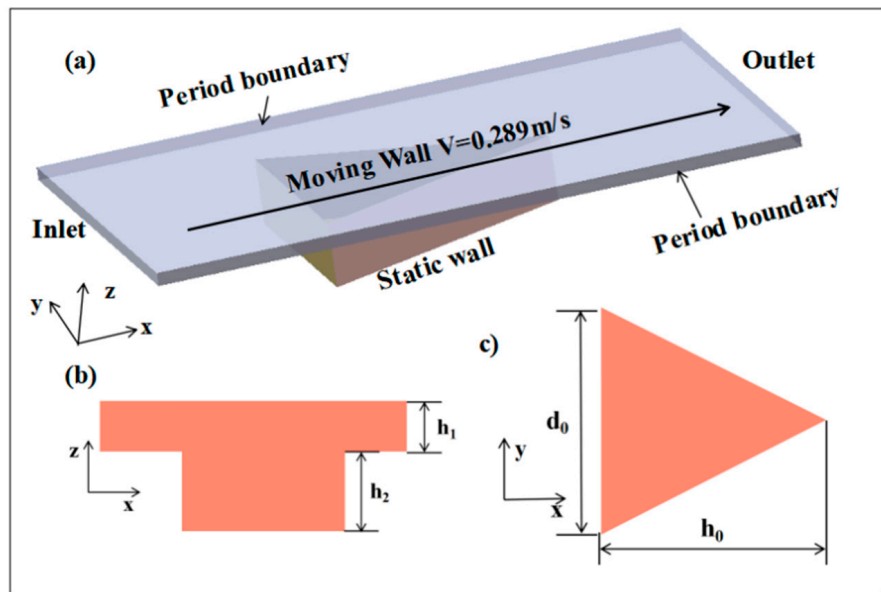

**Figure 4.** (**a**) Boundary conditions of single-texture flow field model. (**b**) Texture size. (**c**) Oil film depth.

We integrated the oil film pressure $p$ of the wall in the area domain to obtain the dimensionless oil film bearing capacity $F_Z$; we integrated the shear force $\tau$ on the upper wall in the area domain to obtain the dimensionless friction resistance $F_x$. Then, Formula (2) is:

$$F_Z = \int \int p \, dxdy$$
$$F_x = \int \int \tau \, dxdy$$
(2)

## 3. Experimental Results and Discussion

### 3.1. Phase Analysis of Sample before and after Nitriding

Figure 5 shows the XRD pattern of the treated 4Cr10Si2Mo. The figure mainly shows $Fe_3O_4$, $Fe_2N$, and $Fe_4N$ diffraction peaks, among which $Fe_3O_4$ and $Fe_2N$ are relatively intense. In the salt bath nitriding and nitriding, cyanate $CNO^-$ decomposes active N atoms. Due to the concentration gradient between the salt bath and the surface, [N] atoms penetrate the surface to form a nitrided layer. The interstitial diffusion mechanism of [N] in Fe nitrides was introduced in [28]. When the nitriding temperature reaches 560 °C, [N] forms the nitriding layers $Fe_4N$ and $Fe_2N$ through interstitial diffusion. Among them, the $Fe_2N$ atomic binding ability is stronger than $Fe_4N$. It can be seen from Figure 5 that the $Fe_2N$ phase strength is higher than that of $Fe_4N$, indicating that the surface corrosion and thermal stability will be further increased after nitriding. Fe and Fe nitrides are fully oxidized to form $Fe_3O_4$ after nitriding, and a black oxide film is formed on the surface, reducing friction and anti-wear during the wear process [29].

### 3.2. Elemental Valence Analysis before and after Nitriding

Figure 6 is the XPS full spectrum of 4Cr10Si2Mo before and after nitriding. Due to the preparation of test specimens, wire cutting was required. After cleaning and drying, the detection errors caused by the cutting fluid cannot be avoided entirely, so the matrix was mixed with elements such as N and O. The C1s peak binding energies before and after nitriding were 285.06 eV and 285.07 eV, respectively. The C1s standard binding energy was 284.8 eV, so the analysis must correct +0.26 eV and +0.27 eV, respectively.

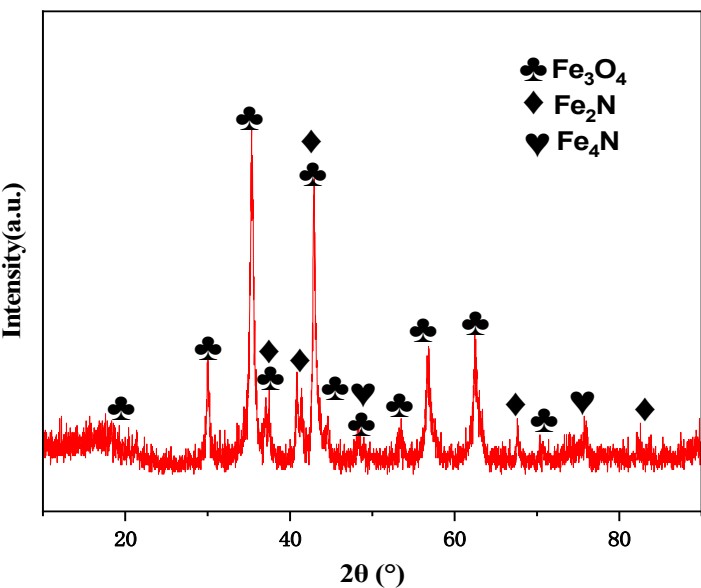

**Figure 5.** Nitriding surface XRD.

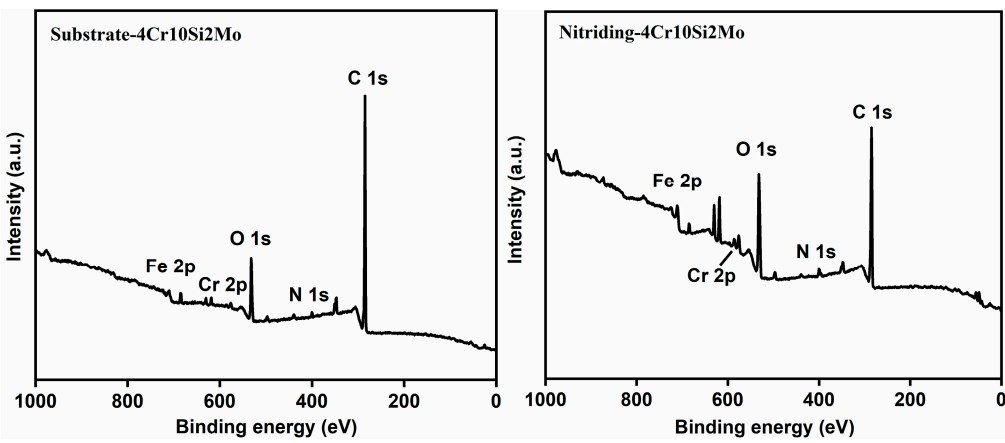

**Figure 6.** XPS full spectra of substrate-4Cr10Si2Mo and nitriding-4Cr10Si2Mo.

　　Figure 7 shows the results of a high-resolution scan performed in a small range of binding energies, which allows for a more precise determination of elemental peak positions. In addition, by fitting the peaks to the corresponding combinations of elements with different chemical states, it is possible to understand the difference in surface chemical properties before and after nitriding. It is not difficult to find from the figure that there are apparent differences in the positions of the peaks of the same element before and after nitriding. Figure 7a shows the fine spectrum of Fe before and after nitriding. Before nitriding, the peaks of Fe2p1/2 and Fe2p3/2 mainly correspond to $Fe^{3+}$, $Fe^{2+}$, and Fe-Fe, while after nitriding, only $Fe^{3+}$ and $Fe^{2+}$ are present, indicating that after nitriding, Fe is oxidized and nitrided, and Fe nitrogen oxides $Fe_2N$, $Fe_4N$, and $Fe_3O_4$ are formed on the surface. After nitriding, the $Fe^{2+}$ peak intensity is much higher than other peaks, which is consistent with the XRD results, and the relative content of $Fe_2N$ is higher than other compounds. After nitriding, the $Fe^{2+}$ peak intensity is much higher than other peaks, which is consistent with the XRD results, and the relative content of $Fe_2N$ is higher than other compounds. The main peak binding energies of $Fe^{3+}$ and $Fe^{2+}$ before nitriding are 711.9 eV and 709.64 eV, respectively, and the corresponding main peak binding energies after nitriding are 713.16 eV and 709.84 eV, respectively. It is found that the binding energy after nitriding shifts to the direction of higher binding energy, indicating that the metastable nitrogen supersaturated solid solution phase transforms into a more stable nitride phase.

In Figure 7b, characteristic peaks appear at 573.66 eV, 575.79 eV, 576.71 eV, etc., before nitriding, and the binding energy mainly corresponds to Cr–Cr and Cr–O bonds. After nitriding, there is only a characteristic peak at 575.77 eV, which corresponds to the Cr–N bond. The intensity of the Cr–N bond after nitriding is much higher than that at each characteristic peak before nitriding. Excluding the influence of pollutants, it can be inferred that after nitriding, the N penetrates the matrix, and CrN is formed under the strong affinity of Cr and N. It is found in Figure 7c that the metal–N bond is formed at 399.9 eV compared with that before nitriding. It can be inferred that the nitriding layer forms Fe nitrides and a small amount of CrN.

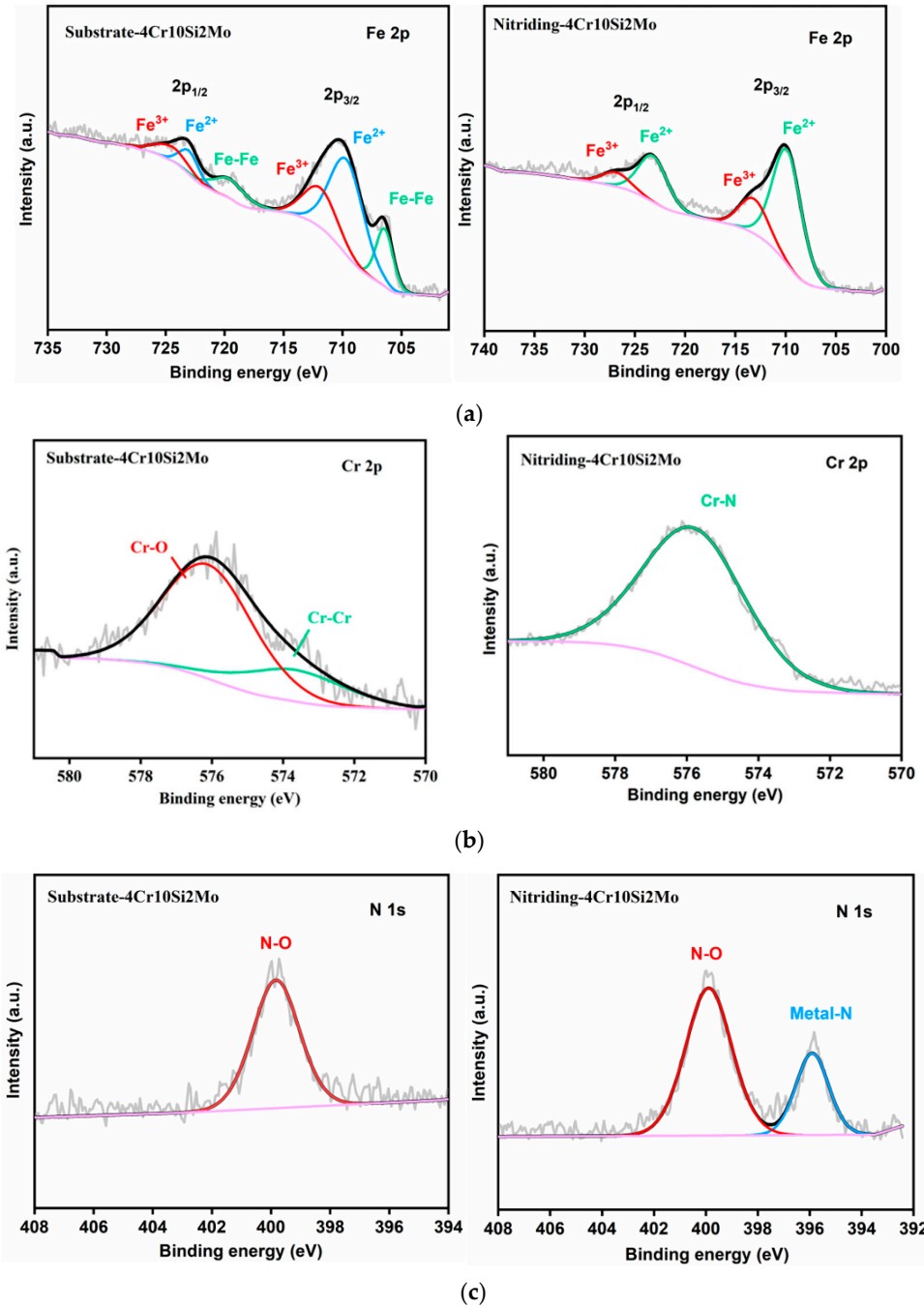

**Figure 7.** Substrate 4Cr10Si2Mo and nitrided 4Cr10Si2Mo element XPS fine spectrum. (**a**) Fine spectrum of surface Fe before and after nitriding. (**b**) Fine spectrum of surface Cr before and after nitriding. (**c**) Fine spectrum of surface N before and after nitriding.

### 3.3. Surface Hardness and Thickness of Nitriding Layer of Nitriding Sample

Table 3 shows the test results of the surface hardness of the substrate and nitrided 4Cr10Si2Mo. The average hardness of the samples before nitriding was 344.2 $HV_{0.1}$, while the average hardness specimens after nitriding reached 710.5 $HV_{0.5}$, and the hardness increased by 51.6%. The main reasons for the increase in hardness are the nitrogen atom matrix diffusion, the formation of nitride, and the matrix solid solution strengthening to enhance the hardness. Moreover, the oxidation film of the surface-generated $Fe_3O_4$ can enhance the surface hardness; nitriding in the generation of a small amount of CrN to enhance the surface hardness also has a catalytic effect. Figure 8 is a cross-sectional view of nitriding. In the figure, the nitriding layer accounts for a large proportion, and the matrix part accounts for a small proportion. The boundary between the nitriding layer and the matrix can also be seen. After measurement, the thickness of the nitriding layer is 240.9 μm.

**Table 3.** Surface hardness of substrate and nitrided 4Cr10Si2Mo (Unit HV).

| Points | 1 | 2 | 3 | 4 | 5 | 6 | Averages |
|---|---|---|---|---|---|---|---|
| Substrate | 327.6 | 326.0 | 339.4 | 349.9 | 358.8 | 363.3 | 344.2 |
| Nitriding | 701.1 | 712.9 | 695.0 | 685.1 | 722.8 | 748.8 | 710.5 |

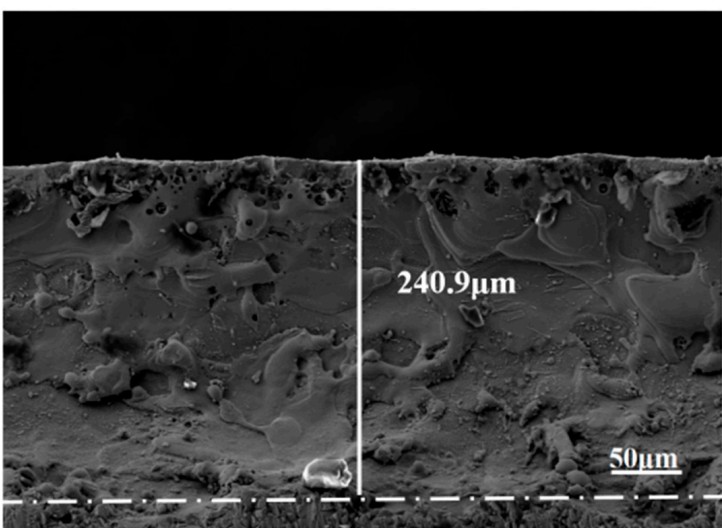

**Figure 8.** Cross-sectional view of nitriding sample.

### 3.4. Analysis of Surface Texture Samples before and after Nitriding

Figure 9a,b shows the whole specimen of nitriding and non-nitriding, and it is obvious that the surface of the non-nitriding specimen is bright white, and the surface of the nitriding specimen is black. Figure 9c shows the textured surface of the substrate 4Cr10Si2Mo. The figure shows that the middle area of the texture is significantly higher than the edge, mainly due to the material splashing caused by repeated high-temperature etching during the laser etching process, which formed a raised state. Figure 9d shows the textured surface of the nitriding sample. There are flaky oxide layers and a few pits on the upper surface. This is a normal phenomenon after salt bath nitriding, and this area is generally called the "loose layer". Matauschek [30] believed that the formation of a "porous layer" on the surface of the nitride layer is due to the Kirkendall effect, where the iron atoms are transferred from the outside to the inside, and the location of lattice defects is shifted from the outside to the inside, leading to the formation of voids. Due to the existence of the loose layer, the surface roughness of the samples increased significantly after nitriding.

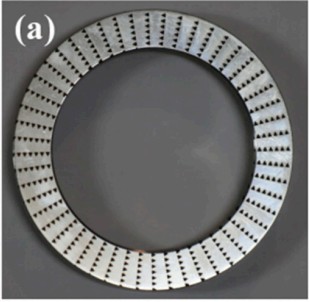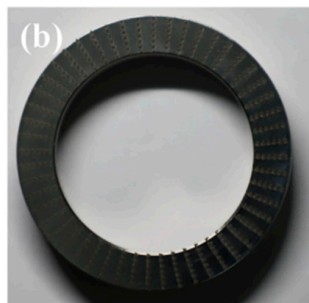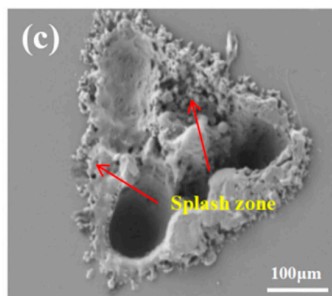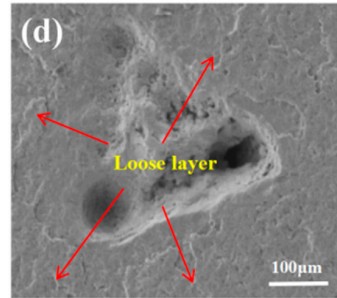

**Figure 9.** Test sample appearance diagram. (**a**) Substrate texture. (**b**) Nitriding texture. (**c**,**d**) are enlarged images of the individual textures in (**a**,**b**).

### 3.5. Analysis of Frictional Wear Mechanism

As shown in Figure 10, the friction coefficient changes under each condition. The first 45 min of the test is the running-in stage, and the last 195 min is the stable wear stage. During the running-in stage, the friction coefficient will rise sharply, which is due to the microscopic and macroscopic geometric defects on the surface of the friction pair, resulting in the initial extreme contact surface being the peak point of the friction pair, and the average surface pressure is considerable [31,32]. According to the friction coefficient variation curve, it can be seen that each nitrided surface enters the stable wear stage significantly faster than the non-nitrided surface, and the friction coefficient of the nitrided surface gradually decreases with time. It shows that the friction stability of the samples after nitriding at room temperature is obviously enhanced. From the friction coefficient curve, it can be seen that the friction coefficient of the overall experimental group is relatively low in the stable phase, which is caused mainly by the good lubrication performance between the friction pairs. The graphite contained in it is a good lubricant, and the small holes left by the graphite exfoliation during movement have the function of adsorbing and storing lubricating oil. Among the non-nitrided samples, the friction coefficient of the GH group is the highest, the friction coefficients of groups TA and TC are significantly lower than those of the GH group, and the friction coefficient of the TB group is close to that of the smooth matrix GH group. The NA group has the most apparent friction reduction effect among the nitriding groups, and its average friction coefficient is only 0.0463. In contrast, the NS group has an average friction coefficient of 0.1033, reducing by 55%. Of course, the NB and NC groups also had friction reduction effects, but the effect was not noticeable. The overall average friction coefficient of all specimens after nitriding was lower than that of non-nitriding. Etsion [33], Yin [34], and Andersson [35] studied the wear of laser-etched textured samples in the presence of oil, and they all concluded that lubricants could reduce the wear of textured surfaces. This is consistent with the experimental results of the present study. This is mainly because the surface texture can increase the thickness of the oil film, thus improving the anti-friction and anti-wear properties. The texture can also store lubricant, and when the lubricant on the material surface is exhausted, the oil stored in the texture can immediately surge out of the surface for secondary lubrication, thus reducing the friction coefficient of the material surface [36]. Many researchers [37–40] believe that the texture area occupancy rate of 10–30% can have good anti-friction efficiency. Hou [41] believed that a textured surface with an occupancy of about 20% had a relatively low coefficient of friction value at different speeds and experimental forces. In this experiment, the area occupancy rates of the triangular texture are 11.45%, 23.86%, and 38.98%. The triangular texture TA and NA groups with an area occupancy rate of 11.45% all show good friction coefficients. The experimental conclusion is contrary to the conclusions of most of the previous researchers. The possible reasons are mainly that this experiment was conducted under the condition of lubricating oil and self-lubricating rubbing pair, which has good lubricating conditions, and the friction coefficient of each group was low. In the experiment, the area occupancy rate was changed by controlling the size of the triangular texture. The

texture with larger bottom and height increased the surface roughness, which led to a larger friction coefficient. In the experiment, the friction coefficients of the NC and TB groups were greater than those of the NS and GH groups.

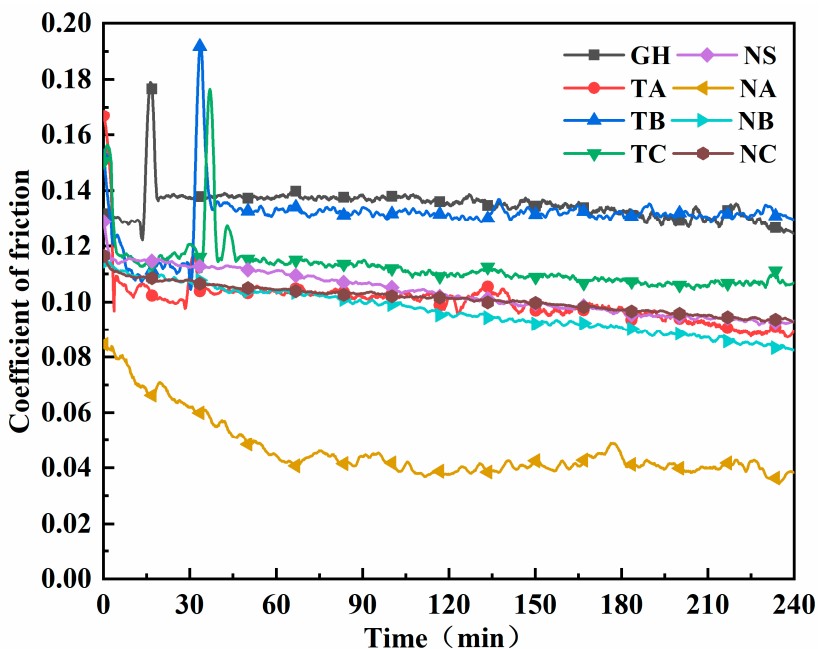

**Figure 10.** Time–friction coefficient curves for all experimental groups.

Figure 11 shows the topography of the worn surface of the sample under an optical microscope before and after nitridation. It can be seen from the figure that the wear of the non-nitrided sample is relatively significant. The surface shows failure and spalling, the width and depth of the wear scar are large, the entire friction surface has obvious furrows, and the wear is very serious; this is because the upper and lower samples are in rigid contact, and the contact pressure is large, which produces a large amount of wear debris, and the fallen wear debris continues to rub the surface, which leads to the generation of surface furrows. Under the repeated action of pressure, the grinding chips are crushed into abrasive particles and the abrasive particles are distributed on the surface, resulting in an uneven surface and accelerating the speed of surface wear. The upper specimen HT200 contains a large amount of flake graphite, which generates concentrated stress at the graphite tip during the reciprocal friction process and is prone to fracture and aggravate the abrasive wear. The surface wear condition of the TA, TB, and TC single texture groups seems to be unsatisfactory, mainly due to the slight difference in hardness between the upper and lower samples, which are severely worn under high contact stresses; a large amount of material is flaked off, and the flaked material collects in the texture cavity, resulting in a gradual shallowing of the texture depth until failure. The wear mechanism of the non-nitrided group is mainly abrasive wear and adhesive wear [42]. Compared with the non-nitrided samples, the wear degree of the nitrided samples is significantly reduced, mainly due to the increase in the surface hardness of the samples after nitriding, and their resistance to plastic deformation and adhesive wear is significantly enhanced [43]. The primary wear mechanism is adhesive wear, manifested as light wear. The surface of the NA, NB, and NC groups showed slight and uniform wear scars, while the NS group was relatively serious, and the black films at both ends of the wear scars had gradually faded away. To summarize, under high pressure for a long time, due to the relatively low hardness of the non-nitrided group, there is a certain degree of wear even under good lubricating performance. In contrast, the wear degree of the nitrided sample is significantly reduced.

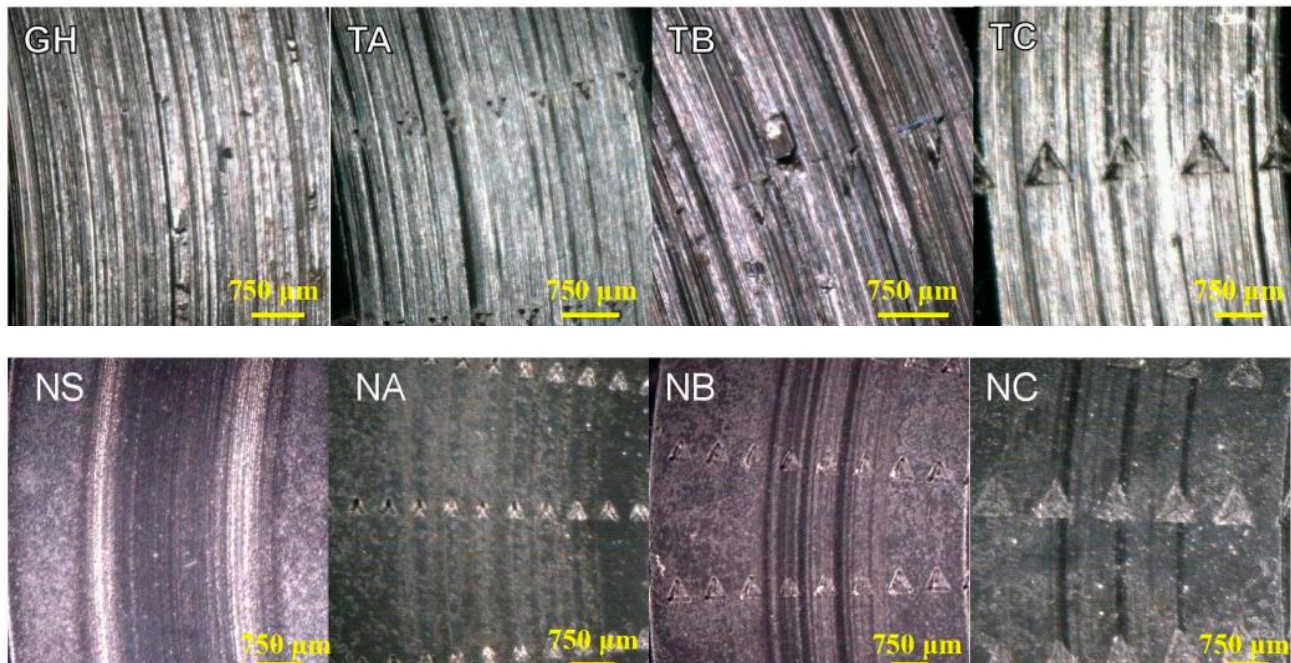

**Figure 11.** Experimental wear morphology of each group under optical microscope.

Figure 12 shows the wear loss for all experimental groups. Because the surface density of the sample after nitriding is different from that of the matrix, when analyzing the wear loss, the non-nitriding group and the nitriding group are compared separately. In the non-nitrided condition, the wear amount of the texture is much lower than that of the smooth specimen. However, it can be seen from Figure 11 that only the textured sample is not evident in terms of wear resistance. Hence, the main reason for the significant difference in wear between the two is that the material is transferred during the friction process, and part of it is stored inside the texture. Under repeated action, it adheres to the interior and surface of the texture. It cannot be dropped even under ultrasonic cleaning, resulting in little difference in the quality of the textured samples before and after wear. Under the nitriding condition, the wear amount of the smooth sample is twice that of the textured sample, so the composite nitriding texture can maintain excellent wear resistance.

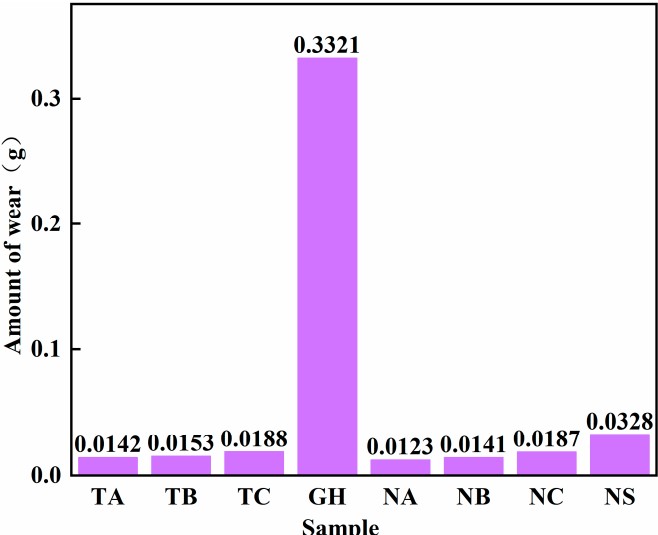

**Figure 12.** Experimental wear loss in each group.

In order to further explore the wear mechanism of 4Cr10Si2Mo valve steel after texture composite nitriding treatment, the micro-surfaces of TA, NS, and NA groups were analyzed. In Figure 13a, it is not difficult to see that the inner cavity of the texture in the TA group is obviously filled with wear debris, and there are also scattered small particles of wear debris on the furrows around the texture. The EDS results showed that a large amount of Fe and Cr elements and a small amount of C and O elements were detected on the worn surface, indicating that an oxidation reaction occurred during the wear process. Among them, the C element is mainly adhered to the surface of the material by black blocks, so it is speculated that the C element comes from HT200. During the wear process, the pressure on the HT200 exceeded its limit value, and plastic deformation occurred. As the wear continued, the HT200 gradually peeled off. The EDS results show that the Wt% of C and O elements at points 2 and 3 at the top corners of the texture are much higher than those at the bottom point 1, and the proportion of wear debris containing C and O elements in the top corners is higher. There may be two reasons for this phenomenon: the first reason is that the concentrated stress at the apex of the triangle increases, and the material wears seriously; the second reason is that the pin rotates along the bottom of the triangle to the apex. The wear debris is accumulated in the top corner with the drive of inertia and lubricating oil. By observing Figure 13a, the remaining texture area at the bottom of the triangular texture is larger than that in the top corner, so the second reason seems more reasonable. Figure 13b,d shows the NS and NA groups. It is not difficult to see that the distribution area of O elements on the surface after wear is significantly larger than that of N elements, so it is believed that the surface of the nitriding group after wear is mainly composed of magnetite $Fe_4O_3$. $Fe_4O_3$ can reduce the coefficient of friction and has stable chemical properties, which can significantly improve the material's wear and corrosion resistance. The ability to resist oxidation during wear is also particularly obvious. The EDS results in Figure 13c show that the Wt% of Cr and N elements at the bottom of the wear scar (points 5 and 6) are significantly larger than those on both sides of the wear scar (points 7 and 8). It can be speculated that when the wear scar reaches a certain depth, the proportion of CrN will gradually increase. In the NA group, there do not seem to be many deep wear scars, so the texture composite nitriding is better than nitriding only in reducing friction and wear resistance.

Figure 14 shows the change curves of the worn surfaces and cross-section topography of TA and NA, respectively. In Figure 14b, the contour of the triangular texture part appeared higher than the non-textured plane area, and the texture is overloaded to collect wear debris. However, it can also prove that the texture efficiently collects abrasive particles and wear debris in friction and wear. In this paper, due to the high applied load, both the upper and lower samples are peeled off under long-term wear, leading to texture overload. In Figure 14b, the worn surface profile of the nitrided textured surface is basically around the 0 scale line, and there is no noticeable fluctuation, indicating that the surface of the nitrided texture remains intact after friction and wear. The three corners of the texture remain intact. The depth range is 30–64.55 μm, the depth of the middle is 14 μm, and the texture depth fluctuates widely. The main reason is that some wear debris from the upper sample pins may collect in the texture, and texture errors are caused by laser processing.

Figure 15a–d shows the three-dimensional topography of the local wear scars of GH, TA, NS, and NA, respectively, among which Figure 15b,d depicts the smooth area away from the triangular texture. Table 4 shows the worn surface roughness characterizations Sa and Sq under different treatments. In Figure 15a, it can be seen that the wear surface of the 4Cr10Si2Mo substrate is densely covered with deep wear scars, and the deepest wear scar reaches 6.451 μm. In Figure 15b, a deep groove appeared at both ends with a depth of 4.109 μm, and other parts also had shallower wear scars. In Figure 15c, it can be seen that the number of wear scars on the plane is small, but the rough contour fluctuates up and down the 0 scales, caused by the pits generated after nitriding. In the NA group in Figure 15d, there is a significantly deeper wear scar; the most profound depth reaches 5.199 μm. In Table 4, the Sa and Sq of the GH group are larger than those of the TA group,

indicating that the existence of texture can reduce the number of wear scars. However, after nitriding, the Sa and Sq of the NA group are larger than those of the NS group. The wear scar is more profound than that of the NS group. It can be concluded that the abrasive and adhesive wear occurred in a part of the material transfer during the wear process. The existence of abrasive particles will make contact between the friction pairs uneven, resulting in severe wear scars. Because the textured surface can store abrasive particles and debris, the number of particles on the surface of the GH and NS groups is more than that of the TA and NA groups. Under the same conditions, the number of surface dents in the GH and NS groups was more than in the TA and NA groups. However, the TA and NA groups may have deeper indentations in the local area. This is because the number of abrasive grains on the textured surface is smaller than that on the smooth surface. Under the same load, these abrasive grains will bear tremendous local pressure, and a small number of deeper wear scars will appear on the surface of the textured sample.

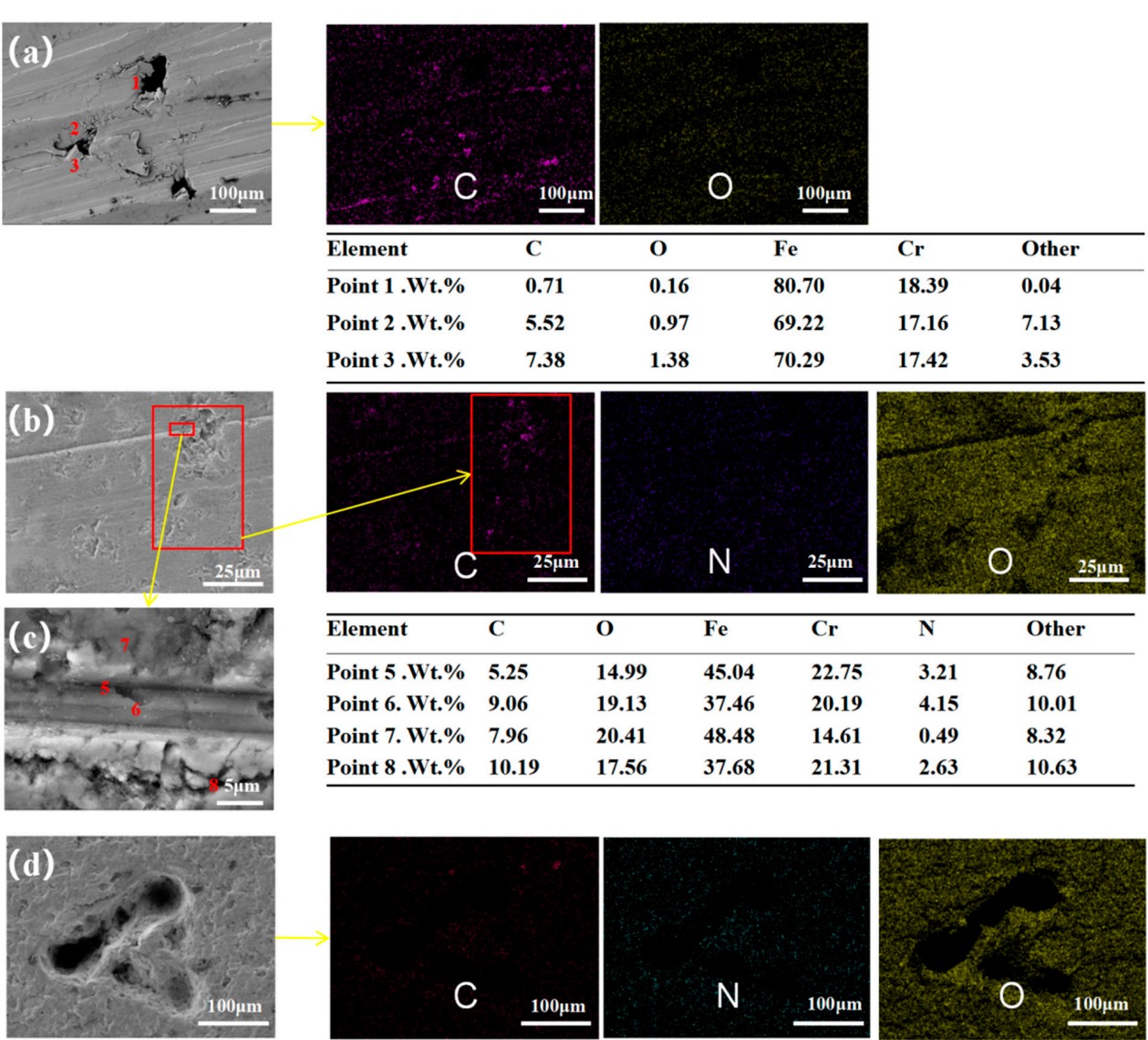

| Element | C | O | Fe | Cr | Other |
|---|---|---|---|---|---|
| Point 1 .Wt.% | 0.71 | 0.16 | 80.70 | 18.39 | 0.04 |
| Point 2 .Wt.% | 5.52 | 0.97 | 69.22 | 17.16 | 7.13 |
| Point 3 .Wt.% | 7.38 | 1.38 | 70.29 | 17.42 | 3.53 |

| Element | C | O | Fe | Cr | N | Other |
|---|---|---|---|---|---|---|
| Point 5 .Wt.% | 5.25 | 14.99 | 45.04 | 22.75 | 3.21 | 8.76 |
| Point 6. Wt.% | 9.06 | 19.13 | 37.46 | 20.19 | 4.15 | 10.01 |
| Point 7. Wt.% | 7.96 | 20.41 | 48.48 | 14.61 | 0.49 | 8.32 |
| Point 8 .Wt.% | 10.19 | 17.56 | 37.68 | 21.31 | 2.63 | 10.63 |

**Figure 13.** SEM image and EDS (point sweep and surface sweep) of the worn part of 4Cr10Si2Mo. (**a**) TA group. (**b**,**c**) NS group. (**d**) NA group.

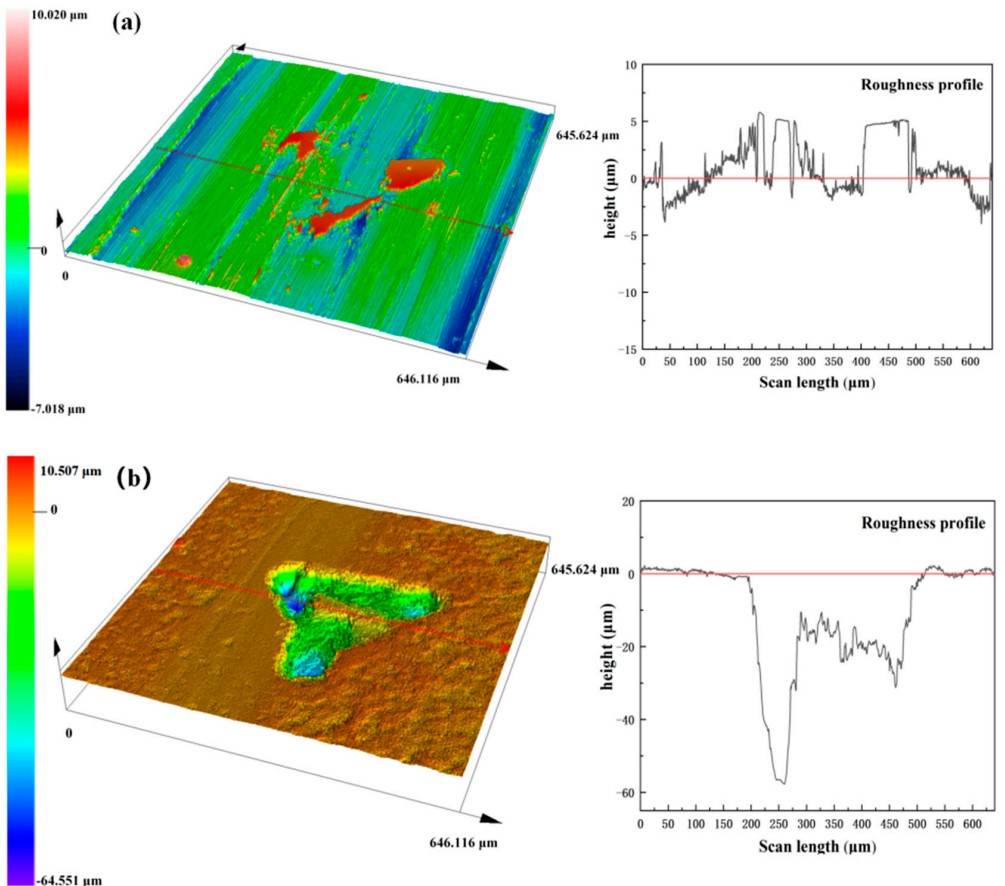

**Figure 14.** Three-dimensional topography of the worn surface of textured samples. (**a**) TA group. (**b**) NA group.

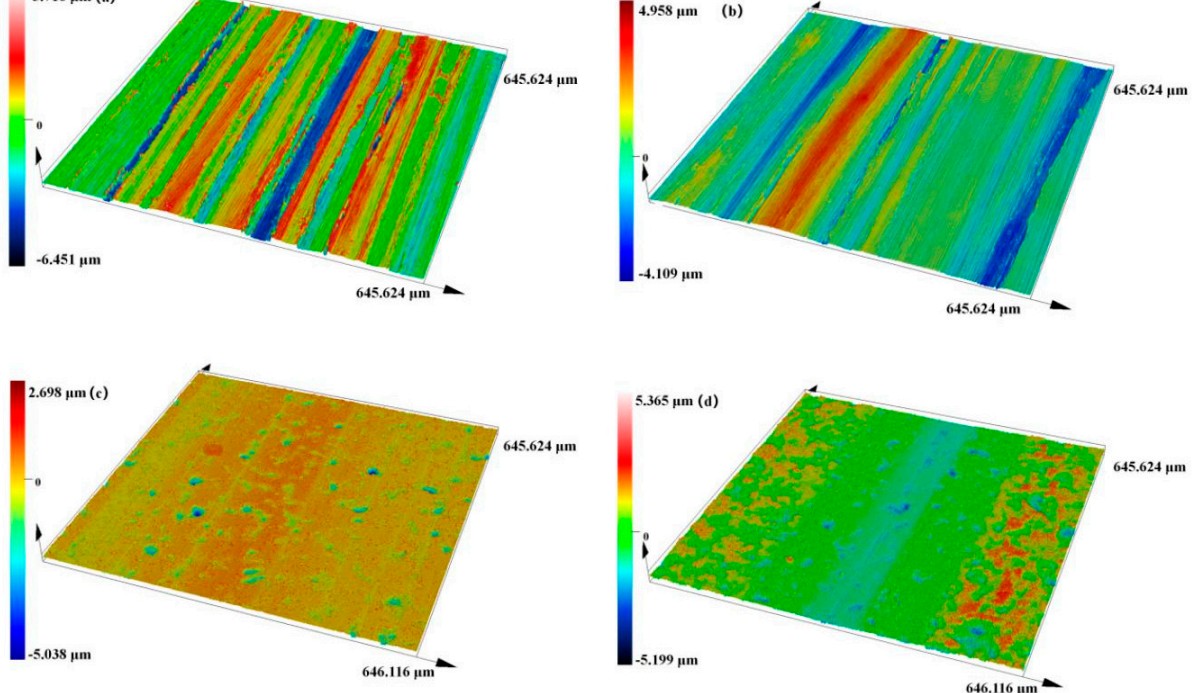

**Figure 15.** Three-dimensional topography of the worn surface. (**a**) GH group. (**b**) TA group unstructured area. (**c**) NS group. (**d**) NA group unstructured area.

**Table 4.** Characterization of wear surface roughness under different treatments Sa and Sq.

| Group | GH Group | TA Group Unstructured Area | NS Group | NA Group Unstructured Area |
|---|---|---|---|---|
| Sa, μm | 1.332 | 0.87 | 0.308 | 0.733 |
| Sq, μm | 1.66 | 1.136 | 0.45 | 0.907 |

## 4. Discussion of Simulation Results

At present, researchers have concluded that texture can improve tribological properties, which mainly have the following six mechanisms: (1) the abrasive particles can be collected during the friction process [44]; (2) the micro-hydrostatic pressure in the texture cavity [45]; (3) the dynamic pressure of the fluid [46]; (4) reduce the effective contact area of the friction pair surface [47]; (5) when lubricated, the textured surface can improve the surface hydrophobicity [48]; (6) store lubricant, and when there is a lack of lubricant, it can be replenished in time for relubrication [48]. This paper uses finite element analysis software to analyze the stress change of textured surfaces and the influence of texture on flow field lubrication.

Figure 16 presents the cloud diagram of the surface stress distribution of the lower sample, in which Figure 16a1–d1 is the texture surface stress distribution cloud diagram of different texture densities at rest. Figure 16a2,b2,c1,d1 is a contour map of surface stress distribution moving to 0.05 s. From the perspective of pressure distribution, apparent stress stratification appeared in the stressed area. The high-stress area was mainly concentrated in the outer ring. It shows that the existence of torque will affect the force distribution on the surface during the rotation process. The stress in the textured region is higher than that around the texture on a textured surface. From the numerical analysis of stress, compared with the peak stress, the peak stress of the textured surface is higher than that of the untextured surface; among the textured surfaces, the surface stress peak of the sample with a texture density of 11.45% is the lowest. Among them, the peak value at rest is $7.1 \times 10^{-3}$ GPa, and the peak value when moving is $8.3 \times 10^{-3}$ GPa. In the inner ring range, the stress on the surface is less than the outer ring surface stress. At rest at 0 s, the stress distribution in the inner ring region of the textured surface is compared with that of the untextured surface, and the minimum stress generated by the textured surface is smaller than that of the untextured surface. Among them, the stress distribution of the untextured surface is [$7.1 \times 10^{-4}$–$1.1 \times 10^{-3}$ GPa]; the texture area density is 11.45%, 23.86%, and 38.98%; and the stress distributions of the inner ring region are [$5.9 \times 10^{-4}$–$1.2 \times 10^{-3}$ GPa], [$6.5 \times 10^{-4}$–$1.3 \times 10^{-3}$ GPa], and [$6.9 \times 10^{-4}$–$1.4 \times 10^{-3}$ GPa]. During motion, the stress on the surface of each group of specimens increases correspondingly compared with the stress on the surface at rest due to frictional resistance during motion.

In summary, introducing texture on the surface of the sample reduces the contact area. It significantly increases the average stress on the surface, but the stress concentration in the outer ring of the untextured surface seems to be more prominent. In the textured surface, the stress of the triangular textured region is significantly larger than that of the smooth region, which can also explain the severe wear phenomenon of the textured region in the experiment.

Figure 17a is the oil film pressure distribution on the upper wall with different area occupancies. It can be seen from the figure that the pressure of the untextured surface increases gradually along the flow direction. However, this pressure value is minimal. On the triangular textured surface, negative pressure is generated at the texture inlet, and the pressure gradient diverges outward in an elliptical shape; a pressure peak appears at the texture outlet, and the pressure gradient converges in an annular shape. Compared with the untextured surface, a noticeable dynamic pressure effect is formed at the triangular texture. This is because the fluid suddenly flows into the left side of the texture, and the lumen volume suddenly increases, dispersing the fluid and causing a sudden pressure drop. From the analysis of the pressure value, the pressure peak generated by the triangular

textured surface is much higher than that of the untextured surface. The oil film pressure peaks produced by textures with different area occupancy rates are also quite different. When the area occupancy rate is 38.98%, the oil film peak value is the largest, and when the area occupancy rate is 11.45%, the oil film peak value is the smallest. Figure 17b shows the central pressure of the oil film curve. Figure 18 is a histogram of dimensionless bearing capacity and frictional resistance. It can be seen that when the area occupancy is 11.45%, the dimensionless bearing capacity is the highest, and the frictional resistance is the lowest. It means that when the area occupancy is 11.45%, its oil film pressure peak center has a large dispersion area to the outside. Even though the oil film pressure peak is the lowest, the oil film carrying capacity is high. This is consistent with the experimental results.

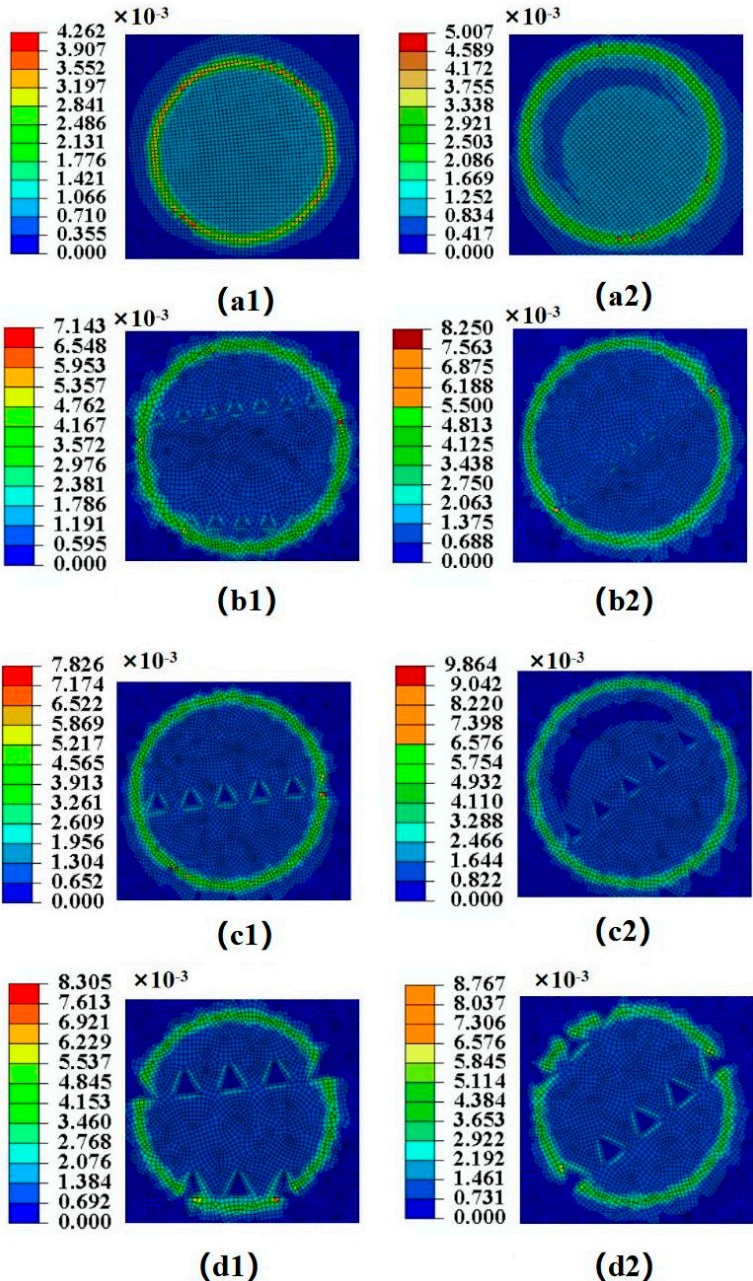

**Figure 16.** Untextured and textured surface stress distribution cloud map at 0 s and 0.05 s (GPa). (**a1**) Ar 0-0 S, (**a2**) Ar 0-0.05 S, (**b1**) Ar 11.45%-0 S, (**b2**) Ar 11.45%-0.05 S, (**c1**) Ar 23.86%-0 S, (**c2**) Ar 23.86%-0.05 S, (**d1**) Ar 38.95%-0 S, (**d2**) Ar 38.95%-0.05 S.

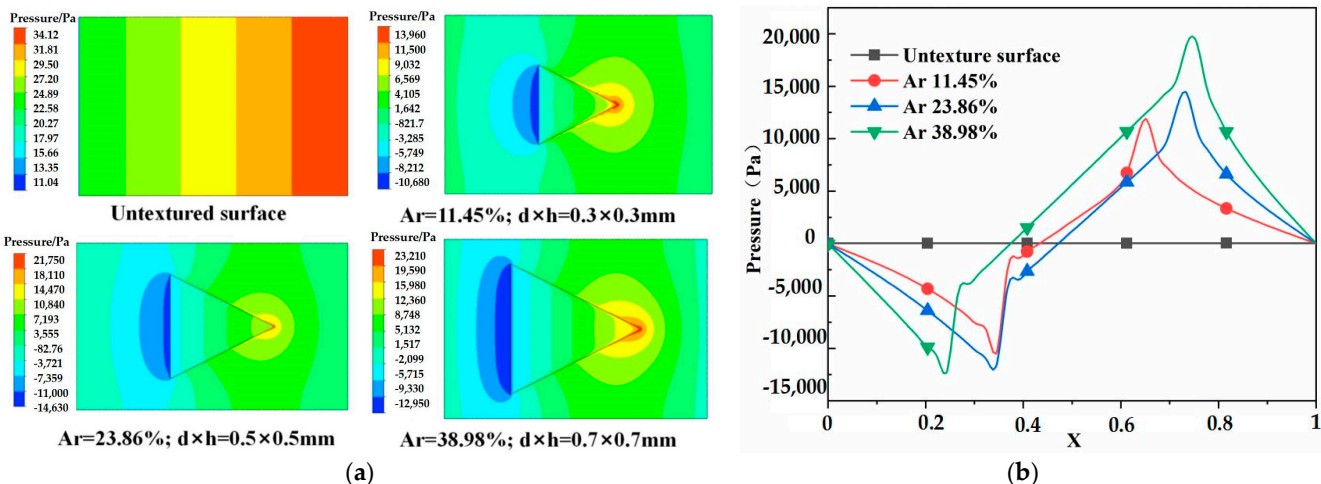

**Figure 17.** Pressure distribution of upper wall and oil film center pressure curve with different texture area occupancies. (**a**) Pressure distribution on the wall surface of the oil film (Pa). (**b**) Oil film center pressure distribution curve.

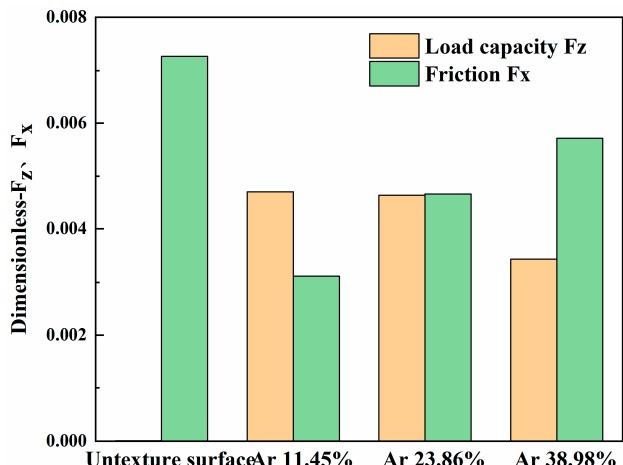

**Figure 18.** The bearing capacity and frictional force of the dimensionless oil film on the upper wall with different texture area occupancies.

From the above, it can be seen that the textured surface in the friction experiment has a significant effect on the friction reduction and anti-wear of the sample. According to the simulation results, the texturing can improve the overall stress concentration on the working surface and play a role in friction and wear reduction. However, the peripheral stress in the triangle of the triangular textured surface is too high compared with other locations. Therefore, the wear near the triangle is more severe during the wear process than in other parts. Figure 19 shows the wear mechanism under texture and composite treatment. Stress concentration occurs at the contact of the outer ring of the pin, the disc, and the texture, which is more likely to be damaged or even fail during friction. In Figure 19a, due to long-term high-load friction, a large number of abrasive particles and debris are collected in the textured cavity, the thickness of the oil film becomes thinner, and the fluid lubrication effect gradually weakens or even fails. On the contrary, in Figure 19b, due to the presence of substances such as $Fe_3O_4$ and $Fe_2N$ in the nitrided layer, the hardness and yield resistance of the material are enhanced, abrasive wear and adhesive wear are reduced, and the texture fluid lubrication effect is optimal. Therefore, the tribological properties of 4Cr10Si2Mo modified by texture compound nitriding are more advantageous.

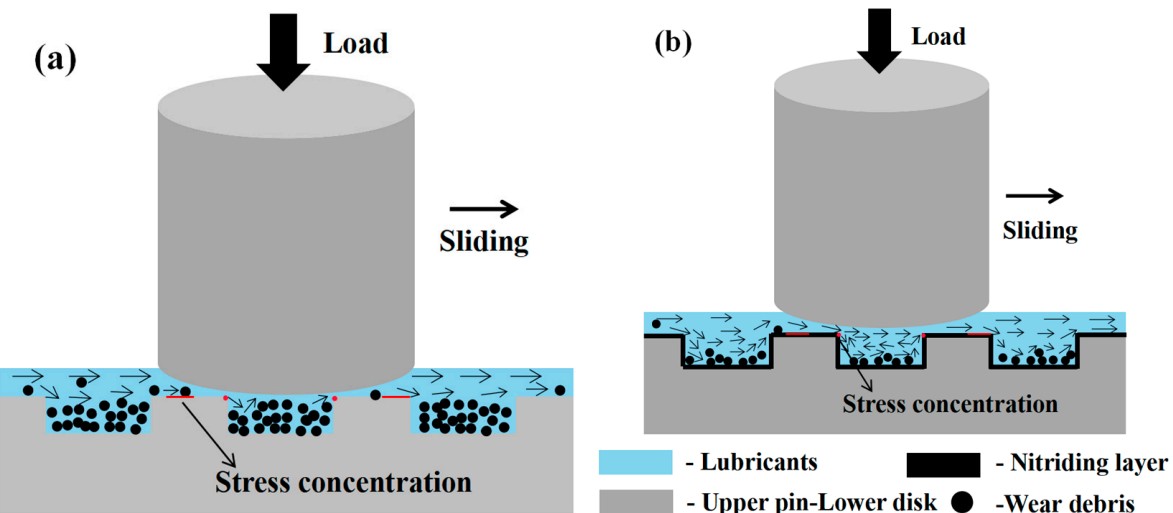

**Figure 19.** Wear mechanism diagram. (**a**) Texture only. (**b**) Texture composite nitriding.

## 5. Conclusions

In summary, a nitride–oxide layer composite texture was formed on the surface of 4Cr10Si2Mo by laser processing and salt bath nitriding and reoxidation treatment. Tribological wear tests and simulation analysis investigated the tribological properties of the nitrided composite textured 4Cr10Si2Mo surface. Based on the experiments and analyses, the following conclusions were drawn.

1. After nitriding and oxidation in the salt bath, the surface hardness of the sample was significantly enhanced, the surface hardness could reach 710 $HV_{0.5}$, and the hardness was doubled. The upper surface of the nitriding sample was mainly $Fe_4O_3$ and $Fe_2N$, and Cr-N was also detected on the surface. However, the magnetite phase ($Fe_3O_4$) and $Fe_2N$ seem to play a more important role in the wear process. $Fe_3O_4$ can reduce the friction coefficient and reduce oxidation wear. $Fe_2N$ can increase surface hardness.

2. A textured surface can reduce the coefficient of friction. However, the material surface wear was high in the absence of nitriding and high load conditions. Adding texture to the surface increased the average stress on the contact surface, especially the stress concentration near the texture. This can also explain the severe wear near the textured area of the non-nitrided sample in the experiment. Among the different area occupancies, the textured surface with an area density of 11.45% experienced less stress. The triangular texture produced a wedge effect in the fluid lubrication, and the pressure difference was generated in the texture's inner cavity to improve the oil film's bearing capacity. The dimensionless oil film had the most significant bearing capacity and the most minor dimensionless friction resistance when the area density was 11.45%. Through various characterization methods, it can be proved that 4Cr10Si2Mo has a pronounced anti-friction effect under the texture composite nitriding treatment. Among them, the triangular texture with an area density of 11.45% (d × h = 0.3 × 0.3) had the best friction reduction effect, and the friction coefficient decreased by 65%.

3. The collection of abrasive particles by triangular texture can significantly reduce abrasive wear damage to the contact surface. However, due to some uncollected abrasive particles on the textured surface, the local contact stress is too large, resulting in deeper wear scars on the non-textured areas of the textured surface. When the stress on the textured surface exceeds its elastic limit, the material will fail under repeated stress cycles. At the same time, the texture function will gradually weaken or even be destroyed. Therefore, in this work, the surface hardness was increased by salt bath nitriding and oxidation, and the role of texture could be better utilized. This composite modification increases the tribological properties of the 4Cr10Si2Mo valve steel, benefiting the engineering applications of valve steel and even auto parts.

**Author Contributions:** Conceptualization, J.Z.; methodology, Z.W.; software, Y.W.; validation, W.G.; formal analysis, Y.D. and D.L.; investigation, Y.D.; resources, Y.W. and Y.M.; data curation, W.G.; writing—original draft preparation, Y.D.; writing—review and editing, Z.T. and W.C.; visualization, Y.D.; supervision, Z.T. and W.C.; project administration, J.Z.; funding acquisition, Z.T.; W.C. All authors have read and agreed to the published version of the manuscript.

**Funding:** This work was financially supported by the High-end Foreign Experts Introduction Project (G2021039004), the National Natural Science Foundation of China (no. 51865053), and the Joint Special Project for Agriculture in Yunnan Province (202101BD070001-051).

**Institutional Review Board Statement:** Not applicable.

**Informed Consent Statement:** Not applicable.

**Data Availability Statement:** The data presented in this study are available on request from the corresponding author.

**Conflicts of Interest:** The authors declare no conflict of interest.

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
