# Peer review of "Effect of Salt Bath Nitriding and Reoxidation Composite Texture on Frictional Properties of Valve Steel 4Cr10Si2Mo"

_coatings, doi:10.3390/coatings13040776_

Round 1

Reviewer 1 Report

In general, the article is interesting, and the information provided by the authors is important.

After the review of the manuscript, I have the following comments.

1. Section 3.1 Phase analysis of sample before and after nitriding

The author declare that “It can be seen from Figure 5(a) that 4Cr10Si2Mo steel mainly has two diffraction peaks of δ-Fe and γ-Fe, among which martensitic iron is the main one”.

However, the peaks show in Figure 5(a) do not match with those described by the authors.

Could the authors clarify wy?

2. In section 3.3. Surface hardness and thickness of nitriding layer of nitriding sample

The authos declare that the hardeness of the layers results in 710.5 N/mm2. On the other hand, in Table 3, they indicate HV as the units of the hardness. Is this correct? Are the units N/mm2 equivalent to HV?

Please clarify it.

3. Page 9, the authors declare a nitride layer thickness of 240.9 µm, which seems to be so large for nitrided steels according to those reported in literature. Did the autors compare the results whit reported for similar steels and similar experimental conditions?

Reviewer 2 Report

The article is interesting and presents interesting results of research on the influence of the method of improving the tribological properties of 4Cr10Si2Mo valve steel. However, in my opinion, the article needs some corrections and clarifications.

1. 1. in the description of the tribological test methodology, it was stated that synthetic oil 15W-40 was used. Question 1: Are you sure it was synthetic oil - the viscosity grade suggests that it was mineral oil. Question 2: Was it engine oil; if so, it contained a number of additives, including lubricity additives of the ZDDP type, strongly affecting the material of the surface layer; this is not explained anywhere by the authors. Question 3: how the oil was introduced into the tested tribological system.

2. The wear mechanism presented in Fig. 19 is not convincing for me and the role of lubricating oil is nowhere to be seen.

3. Conclusions are written in the subjunctive, e.g. "which can make the texture work better". I believe that conclusions should be written in the form of statements well supported by research results.

Reviewer 3 Report

1. In Fig. 11 add a dimensional segment

2. Convert MPa to GPa

3. The texture is not visible in Fig. 3b

4. What is the depth of one texture pattern element? Are they all the same depth?

5. In fig. 9c,d it can be seen that there is an uneven formation of a pattern in depth. Irregularity in depth can lead to different wear. The tribological properties may depend on this. How do the authors ensure uniformity in depth?

6. How does depth affect wear and coefficient of friction?
